# Comprehensive Analysis of BrHMPs Reveals Potential Roles in Abiotic Stress Tolerance and Pollen–Stigma Interaction in *Brassica rapa*

**DOI:** 10.3390/cells12071096

**Published:** 2023-04-06

**Authors:** Lin Yang, Xiaoyu Wu, Shangjia Liu, Lina Zhang, Ting Li, Yunyun Cao, Qiaohong Duan

**Affiliations:** 1State Key Laboratory of Crop Biology, Shandong Agricultural University, Tai’an 271018, China; 2College of Horticulture Science and Engineering, Shandong Agricultural University, Tai’an 271018, China

**Keywords:** *Brassica rapa*, HMP, expression pattern, abiotic stress, reproduction

## Abstract

Heavy metal-associated proteins (HMPs) participate in heavy metal detoxification. Although HMPs have been identified in several plants, no studies to date have identified the HMPs in *Brassica rapa* (*B. rapa*). Here, we identified 85 potential HMPs in *B. rapa* by bioinformatic methods. The promoters of the identified genes contain many elements associated with stress responses, including response to abscisic acid, low-temperature, and methyl jasmonate. The expression levels of *BrHMP14*, *BrHMP16*, *BrHMP32*, *BrHMP41*, and *BrHMP42* were upregulated under Cu^2+^, Cd^2+^, Zn^2+^, and Pb^2+^ stresses. *BrHMP06*, *BrHMP30*, and *BrHMP41* were also significantly upregulated after drought treatment. The transcripts of *BrHMP06* and *BrHMP11* increased mostly under cold stress. After applying salt stress, the expression of *BrHMP02*, *BrHMP16*, and *BrHMP78* was induced. We observed increased *BrHMP36* expression during the self-incompatibility (SI) response and decreased expression in the compatible pollination (CP) response during pollen–stigma interactions. These changes in expression suggest functions for these genes in HMPs include participating in heavy metal transport, detoxification, and response to abiotic stresses, with the potential for functions in sexual reproduction. We found potential co-functional partners of these key players by protein–protein interaction (PPI) analysis and found that some of the predicted protein partners are known to be involved in corresponding stress responses. Finally, phosphorylation investigation revealed many phosphorylation sites in BrHMPs, suggesting post-translational modification may occur during the BrHMP-mediated stress response. This comprehensive analysis provides important clues for the study of the molecular mechanisms of *BrHMP* genes in *B. rapa*, especially for abiotic stress and pollen–stigma interactions.

## 1. Introduction

There is growing awareness of heavy metal pollution with the continued development of industry and agriculture and its impacts on the environment and human health. Various metals such as iron (Fe), cobalt (Co), chromium (Cr), copper (Cu), magnesium (Mg), selenium (Se), manganese (Mn), molybdenum (Mo), and zinc (Zn) are essential biochemical and physiological nutrients for various functions of the body, and inadequate supplies can lead to various deficiencies in humans [1]. The heavy accumulation of non-essential elements such as cadmium (Cd), lead (Pb), and mercury (Hg) not only affects plant growth and development, but these elements can also accumulate in the human body through the food chain and have a negative impact on human health [2].

The primary feedback mechanism in plants after exposure to high concentrations of heavy metals is the production of reactive oxygen species (ROS). The presence of heavy metals as free ions or the reduction of functional groups such as glutathione sulfhydryl groups result in the generation of ROS, which ultimately leads to oxidative damage in plant cells [3]. Many heavy metals can cause the continuous production of ROS in chloroplasts, mitochondria, and peroxisomes, causing stress and oxidation toxicity in plants [4,5]. Yet, there are mechanisms that have evolved in plants to regulate tolerance to heavy metal toxicity in order to resist poisoning, and heavy metal-associated proteins (HMPs) have a critical function in these mechanisms. HMPs include P1B-type ATPase, also known as the heavy metal ATP enzyme (HMA, part of the large P-type ATPase family [6], heavy metal-associated plant proteins (HPPs), heavy metal-associated isoprenylated plant proteins (HIPPs) [7], and ATX1-like [8].

The roles of different types of HMPs in metal transport, detoxification, and abiotic stress processes have been described. AtHMA1 promotes the detoxification of Zn(II) by decreasing the plasmatic zinc content of *Arabidopsis thaliana* [9]. In *A. canescens*, a membrane-bound protein, AcHMA1, encodes a metal tolerance protein that mediates iron scavenging in eukaryotes and plays an essential role in the abiotic stress response [10]. AtHMA2 is involved in Zn/Cd transport in plants [11]. AtHMA5 regulates Cu tolerance in *Arabidopsis* [12]. In *Arabidopsis thaliana*, the overexpression of ATX1 enhances copper tolerance and plays an important role in maintaining copper homeostasis [13]. P(1B)-type ATPase 1 is involved in the regulation of bidirectional interactions between the plastid and the nucleus [14]. The overexpression of *Hordeum vulgare* farnesylated protein 1 (HvFP1), a HIPP of barley, suppresses the induction of ABA-related genes during leaf senescence and drought [15]. HIPPs are not only responsible for the detoxification mechanisms of heavy metals (especially Cd) but are also involved in plant–pathogen interactions and in responses to cold and drought stresses [7]. AtHIPP3 regulates the salicylic acid-dependent pathogen response pathway through its bound zinc and is also implicated in seed and flower development, with particular attention to the fact that the overexpression of HIPP3 delays flowering [16]. In *Solanum lycopersicum*, HIPP7, a possible key site for 1-naphthaleneacetic acid (NAA), co-regulates heavy metal stress [17]. These studies suggest that HMPs participate in heavy metal transport, detoxification, and response to abiotic stresses, with the potential for functions in sexual reproduction.

Chinese cabbage (*Brassica rapa* L. ssp. *pekinensis*) is a Brassicaceae vegetable that is widely cultivated. Heavy metal pollution can significantly affect cabbage growth [18]. Heavy metal gene families have been identified in several crops [19,20,21], but HMP family members in *Brassica rapa* have not been reported. Abundant studies have shown that HMPs are involved in the transport, detoxification, and response to abiotic stresses of heavy metals and suggest a possible role in sexual reproduction. In this study, we aimed to identify BrHMP gene family members at the genome-wide level and to analyze the predicted physicochemical properties and structures of the proteins and the expression patterns of these genes under stresses of heavy metals, abiotic stresses, and self-incompatible pollination. The results of this study further our theoretical understanding of the function of BrHMP genes to elucidate their role in plant response mechanisms to abiotic stresses and the regulation of pollen–stigma interactions.

## 2. Materials and Methods

### 2.1. Identification and Physicochemical Characterization of BrHMPs

Whole genome sequences, gff3 genome annotation data, and HMP amino acid (aa) sequences of *B. rapa*, *A. thaliana*, and *Oryza sativa* were obtained from EnsembPlants (http://plants.ensembl.org/, accessed on 28 September 2022) for download. The prediction of the *B. rapa* HMP (BrHMP) genes in the whole *B. rapa* genome was based on the previously reported *Arabidopsis* HMP gene family ID using the BLASTP program. Protein structures of genes were predicted using the Pfam (http://pfam.xfam.org/, accessed on 20 October 2022) [22] online website, and candidate genes were further screened by analysis of conserved structural domains to finalize the BrHMP genes. The online website Expasy (http://web.expasy.org/, accessed on 20 October 2022) [23] was used to predict the relative molecular mass, theoretical isoelectric point, and other physicochemical properties of the *B. rapa* HMP proteins.

### 2.2. Chromosome Localization, Synteny Analysis, and Phylogenetic Tree Construction

Chromosomal locations of BrHMP family genes were determined by TBtools and covariance with HMP family genes in *A. thaliana* was determined by MCScanX (Guangzhou, China) using default parameters. Synteny was analyzed with the MCScanX (Multiple Collinearity Scan toolkit) and Circos of TBtools. For phylogenetic tree construction, multiple comparisons of the HMA domain of *Brassica rapa* and *Arabidopsis thaliana* [19] were performed using MEGA X (10.0.1). The maximum likelihood (ML) method was used to construct unrooted trees, using 1000 bootstrap replicates.

### 2.3. Conserved Motifs, Subcellular Localization, and GO Analysis

The predicted conservative motifs of BrHMP were performed using the MEME online website. Protein subcellular localization was predicted using the bioinformatics online analysis software PSORT (https://www.genscript.com/wolf-psort.html/, accessed on 21 November 2022) [24]. Heat map generation was performed using TBtools. Data analysis and visualization were conducted using an online platform to plot GO enrichment results as bubble plots (https://www.bioinformatics.com.cn, accessed on 21 November 2022).

### 2.4. Promoter Element Analysis and Expression Analysis

The promoter analysis was performed using the online tool plantCARE (http://bioinformatics.psb.ugent.be/webtools/plantcare/html/, accessed on 21 October 2022) for a 2000 bp sequence upstream of the BrHMP gene [25] with default parameters.

Transcriptome data were obtained from the Brassicaceae Database (http://brassicadb.cn, accessed on 22 October 2022) and were used to analyze the tissue-specific expression of *B. rapa.* The data were normalized as transcripts per million (TPM). Heat maps were generated using the HeatMap plugin of TBtools (v1.108).

### 2.5. Plant Material, Stress and Pollination Treatments

Experiments were conducted using commercial seeds of the variety ‘DHB 848’. The seeds were planted in nutrient soil or nutrient solution and grown in a plant incubator at 25 °C/15 °C, with a light/dark cycle of 16/8 h and a light intensity of 300 mmol/m^2^/s. Seedlings with six leaves of the same growth stage were selected for subsequent experiments. In the heavy metal treatments, plants were treated with 100 µM CdCl_2_, 500 µM Pb(NO_3_)_2_, 100 µM CuSO_4_ and 500 µM ZnSO_4_. In the abiotic stress treatments, 150 mM NaCl was applied to plant leaves to simulate salt stress and 15% PEG 6000 to simulate drought, while control plants were treated with dd H_2_O. For cold stress treatment, plants were exposed to 4 °C. Samples of plants were collected at 2 h, 4 h, 6 h, and 12 h after treatment. Stigmas of *B. rapa* were samples at 5 min, 10 min, and 20 min after SI and CP, and unpollinated (UP) stigmas were collected as controls. Samples were rapidly frozen in liquid nitrogen and stored at −80 °C. Three biological replicates were used per treatment to extract RNA.

### 2.6. Total RNA Extraction and qRT-PCR

Total RNA was extracted using a FastPure® Cell/Tissue Total RNA Isolation Kit V2 (Vazyme Biotech, RC112, Nanjing, China). Reverse transcription was performed using TransGen (AU341-02, Beijing, China) to obtain the cDNA. ChamQ SYBR qPCR master mix (Q711-03, Vazyme, Nanjing, China) was used for qRT-PCR, with BrActin 2 as the reference gene. Data were analyzed using the 2^−∆∆CT^ method [26]. Excel was used to analyze the data, and the specific primer sequences are listed in Appendix A.

### 2.7. Protein–Protein Interaction Networks and Phosphorylation Site Prediction

Predictions of protein-protein interactions were conducted using the STRING online site (http://cn.string-db.org/, accessed on 21 November 2022) [27] selecting *Arabidopsis thaliana* as the organism and obtaining protein network interaction maps (minimum required interaction score = 0.150; other parameters were set by default).

Phosphorylation sites were predicted via the online website and the statistical analysis representation of the data using Excel 2016.

## 3. Results 

### 3.1. Basic Information and Chromosome Distribution Analysis

To analyze the basic characteristics of the BrHMP gene family, we used 55 amino acid sequences of AtHMPs for comparative screening of the genome of *B. rapa*. We isolated 85 BrHMP gene family members and renamed these genes BrHMP01 to BrHMP85 according to their positions on the chromosome. A relevant bioinformatics analysis of them revealed (Appendix A) that the BrHMP protein length ranged from 67 amino acids (BrHMP23) to 1192 amino acids (BrHMP66); protein molecular masses ranged from 7.1 KDa (BrHMP23) to 130.5 KDa (BrHMP66); and protein isoelectric point ranged from 4.65 (BrHMP19) to 10.13 (BrHMP69); and the BrHMP family members include 29 acidic proteins and 56 basic proteins.

We determined the chromosomal location of HMPs, and the results showed that 83 of the 85 members were distributed on all 10 *B. rapa* chromosomes, while BrHMP06 and BrHMP08 were distributed on the chromosome scaffold (Figure 1). The largest number of genes (15) mapped to chromosome A09, followed by 13 on chromosome A02; nine on chromosome A03; seven each on A04 and A07; six each on A01, A08, and A10; and five members on chromosome A05. In addition, by intraspecific covariance analysis, the results showed that there were 21 chromosomal segmental duplications in the BrHMPs (Figure 1), namely, BrHMP02 and BrHMP60, BrHMP03 and BrHMP61, BrHMP04 and BrHMP30, BrHMP10 and BrHMP85, BrHMP11 and BrHMP22, BrHMP11 and BrHMP84, BrHMP16 and BrHMP82, BrHMP14 and BrHMP24, BrHMP14 and BrHMP81, BrHMP19 and BrHMP46, BrHMP20 and BrHMP64, BrHMP22 and BrHMP84, BrHMP23 and BrHMP41, BrHMP24 and BrHMP81, BrHMP26 and BrHMP36, BrHMP26 and BrHMP39, BrHMP31 and BrHMP75, BrHMP35 and BrHMP56, BrHMP35 and BrHMP74, BrHMP36 and BrHMP39, and BrHMP37 and BrHMP38, respectively. This may have caused the expansion of the BrHMP gene in the *B. rapa* genome.

### 3.2. Phylogenetic Tree and Synteny Analysis

To investigate the HMP evolutionary relationships, we constructed a maximum likelihood phylogenetic tree of HMPs in *A. thaliana* and *B. rapa*. As shown in Figure 2A, based on the evolutionary relationships, we classified the HMPs into six subfamilies, namely, ‘I’, ‘II’, ‘III’, ‘IV’, ‘V’, and ‘VI’. ‘I’ subfamily is the ‘P1B-ATP’, which includes seven members in the BrHMPs. In addition, ‘II’ subfamily is the ‘CCH’, which belongs to the CCH protein family and is involved in intracellular copper homeostasis by transporting and secreting metals and includes 20 members in BrHMPs. The other four subfamilies ‘III’, ‘IV’, ‘V’, and ‘VI’ include 19, 11, 20, and 8 members, respectively.

To further investigate the homology of the *BrHMP* genes, the syntenic relationships with the HMP gene families of *A. thaliana* and *O. sativa* were analyzed (Figure 2B). The results showed a total of 96 pairs of orthologous genes of *HMPs* in *B. rapa* and *A. thaliana* and only three pairs of orthologous genes of *HMPs* in *B. rapa* and *O. sativa*. Thus, there was greater syntenic conservation between *B. rapa* and *A. thaliana* than between *B. rapa* and *O. sativa*. These results suggest that *B. rapa* and *A. thaliana* are more closely related and may be more similar in function.

### 3.3. Subcellular Localization Analysis

To determine the possible sites where BrHMPs perform their functions, we performed subcellular localization analysis. As shown in Figure 3, most of the BrHMP proteins were predicted to localize to chloroplasts, cytoplasm, nucleus, and mitochondria. BrHMP65, BrHMP63, BrHMP41, BrHMP09, BrHMP69, BrHMP74, BrHMP61, BrHMP45, BrHMP23, BrHMP37, BrHMP56, BrHMP50, and BrHMP19 were predicted to be in chloroplasts; BrHMP67, BrHMP17, BrHMP85, BrHMP10, and BrHMP71 localized to the nucleus; BrHMP52, BrHMP75, BrHMP59, and BrHMP54 were predicted to be in the cytoplasm; and BrHMP30, BrHMP08, BrHMP04, and BrHMP05 were predicted to be in the mitochondria.

### 3.4. Gene Structure and Conserved Motif Analysis 

To explore the potential protein functions of BrHMPs, ten conserved motifs of the BrHMP gene, i.e., motifs from 1 to 10, were predicted (Figure 4A,B). Motif 1 and motif 5 were identified as HMA (heavy metal associated domain) structural domains (accession: cd00371). Among the 85 BrHMPs, 80 members contained motif 1 and 25 members contained motif 5, with variation in the composition of other motifs. The HMA structural domain was conserved. It is composed of ~30 amino acid residues and contains two cysteine residues that bind and transfer heavy metal ions such as cadmium, zinc, cobalt and copper [28]. Genes containing the HMA structural domain may function in abiotic stresses, plant development, and immunity [17,29,30]. Motif 8 and motif 10 were identified as HAD (haloacid dehalogenase) structural domains (accession: cl21460). The HAD superfamily catalytic domain active site scaffold consists of a four-loop platform [31], and most HAD proteins are involved in phosphorylation, including phosphatases, P-type ATPases, phosphatases, and phosphotransferases [32]. HAD phosphatases are equipped with a cap structure that controls the active site, and cap-bearing HAD phosphatases can affect the phosphorylation of proteins [33].

The number of exons of *BrHMP* family members varies from 1 to 17 (Figure 4C). Most homologous genes tend to have similar exon–intron structures and transcript lengths. For example, the *AT1g22290* homologs *BrHMP53* and *BrHMP72* each contain two introns and three exons, and the transcript lengths are both close to 500 bp.

### 3.5. GO Analysis 

To further explore the function of the BrHMP gene family, GO analysis was performed, as shown in Figure 5, to classify potential functions into GO categories such as BP (biological processes), CC (cellular components), and MF (molecular functions). The enriched biological processes were mainly related to metal ion transport, cation transport, and establishment of localization. The enriched cellular component category included functions related to membrane composition. The enriched molecular functions were mainly related to the binding and conjugation of metal ions, ATPase activity, and coupling across the membrane. Overall, the results suggest that the main function of BrHMPs is to be involved in metal ion transport with possible involvement in phosphorylation.

### 3.6. Promoter Cis-Regulatory Elements Analysis

To explore the mechanism of BrHMP gene family-mediated response, we performed *cis*-element prediction by analyzing the 2000 bp DNA sequences upstream of the *BrHMP* genes (Figure 6 and Appendix A). By predicting the promoter region of the BrHMP gene, a total of 1890 *cis*-elements was identified, including 983 (52%) associated with growth and developmental responses. There were 544 phytohormone response-related elements, or 28.8%, and 363 stress response-related elements, or 19.2%. Among the *cis*-elements associated with growth and developmental responses, the largest proportion was light responsive elements, at 87.8%. Among the *cis*-elements associated with phytohormone responses, the largest proportion was MeJA-responsiveness, at 42.1%, followed by abscisic acid responsiveness, at 26.5%. Among the *cis*-elements associated with stress response, 57% were related to anaerobic induction, 16.3% were related to low-temperature responsiveness, and 11% were related to drought-inducibility. These results suggest that the BrHMP proteins act in processes such as photosynthesis, growth and development, and response to stress.

### 3.7. Gene Expression Analysis 

#### 3.7.1. Expression Patterns of the BrHMPs in Different Tissues

In order to investigate the expression of *BrHMPs*, we analyzed the expression levels of the 85 *BrHMPs* in callus, flowers, leaf, roots, silique, and stems. Figure 7 and Appendix A show that the pattern of gene expression levels differed between tissues, with more highly expressed genes in roots, flowers, and silique; for example, *BrHMP32*, *BrHMP11*, *BrHMP16*, *BrHMP42*, and *BrHMP78* were highly expressed in roots (TPM>100); *BrHMP08*, *BrHMP16*, *BrHMP78*, *BrHMP02*, *BrHMP32*, and *BrHMP44* were highly expressed in flowers (TPM>100); and *BrHMP11*, *BrHMP41*, *BrHMP78*, *BrHMP36*, *BrHMP14*, and *BrHMP73* were highly expressed in the silique. Genes with higher expression in leaves included *BrHMP06*, *BrHMP14*, and *BrHMP42*, and genes with higher expression in stems included *BrHMP16*, *BrHMP32*, *BrHMP82*, *BrHMP13*, *BrHMP42*, *BrHMP30*, *BrHMP14*, and *BrHMP78*.

#### 3.7.2. Expression Patterns of the BrHMPs in Response to Heavy Metal Stress

HMPs primarily act in the transport of metal ions, so we next investigated the response of *BrHMPs* to heavy metal stress. To do this, we treated *B. rapa* seedlings with Cu^2+^, Cd^2+^, Zn^2+^, and Pb^2+^ to simulate heavy metal stress and measured gene expression in the treated seedlings. Based on the results of expression patterns in different tissues (Appendix A), we selected eight *BrHMPs* with high expression in roots and leaves for qRT-PCR analysis: *BrHMP06, BrHMP11, BrHMP14*, *BrHMP16*, *BrHMP32*, *BrHMP41*, *BrHMP42*, and *BrHMP78*. Compared to no treatment, after 2 h of Zn^2+^ stress treatment, the relative expression levels of six of the tested *BrHMPs* were significantly increased by more than 3-fold, but there was little change in expression for *BrHMP11* and *BrHMP78* (Figure 8A). Under Cd^2+^ stress, there was a significant more than 3-fold increase in the relative expression levels of all BrHMPs, except *BrHMP11*, compared to no treatment (Figure 8B). Under Cu^2+^ stress treatment, the relative expression of *BrHMP14*, *BrHMP16*, *BrHMP32*, *BrHMP41*, and *BrHMP42* increased significantly, more than 3-fold, compared to no treatment; the expression of *BrHMP11* increased nearly 2-fold, while the expression of *BrHMP06* and *BrHMP78* decreased (Figure 8C). Under Pb^2+^ stress treatment, the relative expression levels of *BrHMP06*, *BrHMP14*, *BrHMP16*, *BrHMP32*, *BrHMP41*, and *BrHMP42* were all significantly increased by more than three-fold compared to the no treatment control, while the relative expression levels of *BrHMP11* and *BrHMP78* were all significantly decreased (Figure 8D). The results suggest that *BrHMP14*, *BrHMP16*, *BrHMP32*, *BrHMP41*, and *BrHMP42* function to resist Cu^2+^, Cd^2+^, Zn^2+^, and Pb^2+^ stresses.

#### 3.7.3. Expression Patterns of the BrHMPs in Abiotic Stress 

Prior studies have shown that the HMP gene family is implicated in the regulation of abiotic stress [7,16], and to further investigate the response of the BrHMP gene family to various stresses, we selected 15 *BrHMP* genes highly expressed in different tissues of *B. rapa* and measured changes in expression in response to abiotic stress. We tested three abiotic stress treatments, including treatment with PEG6000 to simulate drought, treatment at 4 °C to simulate low temperature, and exposure to NaCl to simulate salt stress. We assessed the expression by qRT-PCR, and the results showed (Figure 9) changes in the expression patterns of *BrHMPs* under different abiotic stresses.

Under PEG6000 treatment, the expression levels of BrHMP06, BrHMP30, BrHMP32, BrHMP41, BrHMP44, BrHMP63, and BrHMP78 were significantly increased in comparison to the control, with BrHMP06, BrHMP30, and BrHMP41 showing a more than 5-fold increase in expression levels as compared to the control. BrHMP32 and BrHMP78 increased significantly after 2 h of treatment, but the other genes showed the most dramatic changes in expressing levels after 4 h of treatment. The levels of expression of all these genes showed a decreasing trend after longer treatments, with some genes decreasing to expression levels below those of the control (Figure 9A).

Under cold stress treatment, the expression levels of *BrHMP06*, *BrHMP08*, *BrHMP11*, *BrHMP14*, *BrHMP78*, and *BrHMP82* were significantly elevated compared to the control group, but the overall levels of expression were lower than those observed for PEG6000 and NaCl treatments. The expression levels of *BrHMP06* and *BrHMP11* changed most significantly, but less than three-fold. The expression levels of *BrHMP06*, *BrHMP08*, *BrHMP11*, and *BrHMP78* were significantly elevated after 2 h of cold stress treatment compared with the control, and these levels were maintained until 4 h of cold stress treatment. However, the expression level of *BrHMP82* was significantly different only at 12 h of cold stress treatment (Figure 9B).

Compared with PEG6000 and cold stress treatments, the expression levels of all 15 *BrHMPs* showed increasing trends after NaCl treatment at different times compared to the control (Figure 9C), with more than 5-fold increases in the expression levels of BrHMP02, BrHMP16, and BrHMP78, implying that more BrHMPs were involved in the regulation of Na^+^ transport.

#### 3.7.4. Expression of BrHMPs during SI and CP Responses

ROS is involved in pollen–stigma interactions [34], similar to the defense mechanism prompted by pathogenic bacterial invasion [35]. In *B. rapa,* there is increased ROS of the stigma after SI and lower ROS after CP [36]. *AtHMPs* are involved in regulating flower development [16], but a role for *BrHMPs* in pollen–stigma interactions has not been reported previously.

We selected four *BrHMPs* with elevated expression levels in SI and decreased expression levels in CP as candidate genes based on transcriptome data (Appendix A), and then we verified the changes in expression levels by qRT-PCR (Figure 10). The results indicated that *BrHMP36* expression began to increase to about 1.7-fold after 5 min of SI and then decreased after 10 min and 20 min. A decreasing trend was observed at 5, 10 and 20 min of CP compared to the UP. The results suggest that *HMP36* may function in SI-induced ROS.

### 3.8. Protein–Protein Interaction (PPI) Networks 

Identifying protein interactions can elucidate how proteins perform their functions. BrHMPs are closely related to AtHMPs, whose functions have been partially reported. Therefore, we used PPI to predict protein interactions to provide functional insights into each of these proteins (Figure 11 and Appendix A). BrHMP41 expression was most dramatically up-regulated in Cu^2+^ and Cd^2+^ stress conditions, and the analysis revealed (Figure 11A) RAN1, COPT5, HMA6, HMA5, and HMA1 as functional partners of the homologous gene AT1G66240, genes that have all been reported to be involved in Cu^2+^ transport [8,37,38,39,40]. HMA1 is also involved in Cd^2+^ transport [41], so the identification of these potential interaction partners suggests these proteins participate in the detoxification of Cu^2+^ and Cd^2+^.

We predicted the functional partners of AT3G05220 (homologs of BrHMP06 and BrHMP42, the two genes that showed the most significant increases in expression under cold stress and zinc stress, respectively) (Figure 11B). One potential interaction partner is BTS (AT3G18290, zinc finger protein-like protein), whose mutation greatly improves zinc tolerance in *Arabidopsis thaliana* [42]. Additionally, BTS interacts with VOZ1 and VOZ2, which have negative regulatory effects on plant responses to drought and cold stress [43]. This result indicates that AT3G05220 may interact with BTS to play a role in response to drought, cold, and possibly other abiotic stress responses by promoting the degradation of transcription factors VOZ1/2. 

*BrHMP16* expression was most significantly upregulated under salt stress conditions, and the predicted interaction partners of its homologous gene AT3G06130 include CSP3 and CSDP1 (Figure 11C). The overexpression of AtCSP3 improved tolerance to salt stress [44], implying that a possible interaction between AT3G06130 and AtCSP3 helps mediate resistance to salt stress.

*BrHMP36* was identified as a candidate gene that changed expression levels in SI and CP, and two functional partners, TL29 and CHX18, were identified based on the PPI of homolog AT2G36950 (Figure 11D). ROS are involved in *B. rapa* pollen–stigma interactions and self-incompatibility responses, and TL29 was reported to be involved in ROS scavenging [45]. In addition, the mutation of another functional partner, CHX18, affects pollen wall formation [46], suggesting potential function in pollen–stigma interactions.

### 3.9. Phosphorylation Site Prediction

In *Arabidopsis*, HMA8 and HMA6 are p-ATPases [47] that use energy provided by ATP hydrolysis, and phosphorylation modifications occur during Cu^+^ transport [48]. To investigate the phosphorylation modifications of HMP proteins in *B. rapa*, we next predicted phosphorylation sites in these gene products (Figure 12). There were 2213 predicted sites for phosphorylation of BrHMPs, with serine residues accounting for about 62%, threonine residues accounting for 28% of sites, and tyrosine accounting for only 10% of phosphorylation sites. Among the 85 BrHMP genes family members, BrHMP03, BrHMP34, BrHMP49, BrHMP66, BrHMP70, BrHMP80, and BrHMP85 had more than 100 phosphorylation sites. Based on the predicted conserved motifs shown in Figure 4, BrHMP03, BrHMP49, BrHMP66, BrHMP70, and BrHMP80 all contain motif 10, the HAD (halogen dehalogenase) structural domain that affects protein phosphorylation, indicating that these proteins may be more likely to undergo phosphorylation modifications.

## 4. Discussion

Heavy metal-associated proteins (HMPs) are involved in heavy metal detoxification, but there are no reports on HMPs and their functions in *Brassica rapa*. Here, we identified 85 HMPs in *Brassica rapa* and analyzed their physicochemical properties and protein structures, as well as their expression patterns in response to heavy metals, abiotic stresses, and SI/CP pollination. Our results from this study will improve our understanding of BrHMP genes and the role they play in the mechanisms of response to abiotic stresses in plants and in pollen and stigma regulation.

A duplication was observed in all plant genomes that have now been sequenced [49]. Intraspecific covariance analysis revealed that the BrHMP gene has a duplication of 21 chromosomal segments, which may have caused the expansion of the BrHMP gene in the *B. rapa* genome (Figure 1A). *Arabidopsis thaliana* underwent three paleoploid events [50]. A genome triploidy event occurred between 13 and 17 million years ago in the ancestors of Brassicaceae. Changes in the number of gene families in the genome may have contributed to the remarkable morpho-plasticity of Brassicaceae. More than one million synthetic genes have been identified from all protein-coding genes, of which 1052,853 were synthesized from *A. thaliana* genes [51]. According to our synthetic analysis, *B. rapa* and *A. thaliana* have 96 pairs of orthologous genes in HMPs, whereas *B. rapa* and *O. sativa* have only three pairs of orthologous genes in HMPs (Figure 2). This suggests that *B. rapa* and *A. thaliana* are more closely related and may have more similar functions.

In addition to ROS, reactive nitrogen species (RNS) are also produced during stress responses and regulate many biological activities and physiological functions involving plant defense and development [52]. Related studies have shown that NO-responsive HMA domain-containing genes function in plant development and immunity in *A. thaliana* [29]. The structure of a protein determines its biological function. By analysis of its structure and conserved motifs (Figure 4), the results showed that these genes all encode HMA structural domains of ~30 amino acid residues and contain two cysteine residues that bind and transfer Cu, Cd, Co, Zn, and other heavy metal ions [28,32]. HMA domain genes are essential for the transport of metal ions bound to various inhibitors and cofactors in the space–time of cells [29], and proteins with these structural domains perform critical roles in plant growth and pressure sensitivity [7,53].

Mature proteins are transported inside specific organelles in order to perform a stable biological function. Among the results of subcellular localization, on the one hand we found that most of the BrHMP proteins were predicted to be located in chloroplasts (Figure 3), and interestingly, the largest proportion of their *cis*-elements related to growth and developmental responses was light-responsive components, accounting for 87.8% of the results (Figure 6). In addition, GO analysis of BrHMPs showed that the biological processes of enrichment are mainly related to the transport of metal ions, the transport of cations, and the establishment of localization (Figure 5). Therefore, we hypothesize that BrHMPs may co-operate with photosynthesis in the detoxification of heavy metals to regulate plant growth and development. On the other hand, the diversity of subcellular localization of BrHMPs suggests that they may be involved in a variety of biological processes. In addition, the results of the subcellular localization of BrHMPs could provide information for future protein–protein interactions as well as structural and functional aspects of proteins.

Related studies have shown that MeJA alleviated PEG6000 stress on seed germination and seedling growth in rice [54] and improved drought tolerance in *Impatiens walleriana* [55]. In this study, among the *cis*-elements associated with phytohormone responses, the largest proportion was MeJA-responsiveness, at 42.1% (Figure 6 and Appendix A). In combination with our gene expression of BrHMPs under PEG6000 stress (Figure 9A), the expression of *BrHMP30*, *BrHMP41*, and *BrHMP78* was significantly upregulated and predicted a relatively high number of MeJA-responsive *cis*-elements; therefore, they may be involved in the regulatory role of the MeJA response to drought.

Furthermore, salt stress response mechanisms are also regulated by MeJA [54]. For instance, elevated endogenous MeJA synthesis by salt stress promotes salt-induced leaf senescence in *R. trigyna* by regulating RtRbohE and enhancing ROS accumulation [56]. According to our results of gene expression of BrHMPs under NaCl stress (Figure 9C), the expression of *BrHMP02*, *BrHMP16*, and *BrHMP78* was significantly up-regulated, but only *BrHMP78* predicted more MeJA-responsive *cis*-elements (Figure 6 and Appendix A), so it may be involved in the regulatory role of the MeJA response to salt stress.

*BrHMP42* expression increased six- to seven-fold under Cu^2+^ and Zn^2+^ stress conditions and four-fold under Cd^2+^ stress (Figure 8), and *BrHMP06* expression increased six-fold under PEG6000-simulated drought stress and 2.5-fold under cold stress and salt stress (Figure 9). BrHMP42 and BrHMP06 share a common Arabidopsis homolog (AT3G05220)*,* and PPI analysis indicate that functional partners include BTS and ABCB15 (AT3G28345)-ABC transporter protein family proteins (Figure 11B). A number of ABC genes are contained in plant genomes, with over 120 members identified in Arabidopsis [57]. Several ABC transporters are essential for the efflux of toxic molecules, particularly the ABCB subfamily of proteins. Tolerance to cadmium, mercury, and arsenic is also conferred by these proteins and are transported to the vesicles in conjunction with phytosphingosine [58,59,60]. ABC transporters are also involved in the transport of various hormones such as jasmonic acid, abscisic acid, strigolactone, auxin, and cytokinin, while *AtABCB15* is involved in the transport of indole-3-acetic acid (IAA) and 1-naphthylacetic acid (NAA) [61]. It is essential that the plant hormones abscisic acid (ABA) and 1-naphthalene acetic acid (NAA) are present in plants in response to various abiotic stresses [62,63], including drought, cold, heat, and salinity. Thus, when plants are exposed to heavy metal stress, the ABC transporters may help transport metal ions to alleviate toxicity and also protect against damage by regulating endogenous hormone levels. 

Studying the tissue-specific expression of genes helps to further explore the functions they exercise in life activities. In the tissue-specific expression analysis of the BrHMP family (Figure 7), nearly a quarter of the members were expressed in roots, making up the major part, suggesting that the root system plays an important role in the uptake of heavy metal ions and their transport. The next approximately 1/6th of the members are expressed in flowers, and interestingly, it has been discussed that HMPs may play a key role in pollen and become the focus of future gene function studies [19]. Therefore, we conducted a preliminary study on cabbage under SI/CP pollination conditions.

The function of the HMP gene family proteins in sexual reproduction is little known. Self-incompatibility is an important mechanism that arose during the evolution of flowering plants to prevent self-fertilization and promote distant lineage reproduction [64]. *Brassica rapa* is a typical vegetable crop with sporophytic self-incompatibility. Self-compatibility in *B. rapa* is mediated by FERONIA receptor kinase-regulated ROS [36]. The ROS accumulated on the stigma began to increase at 5 min after pollination with ‘SI’ pollen, while at 5 min after pollination with compatible pollen, the ROS accumulated on the stigma began to decrease. ROS in the stigma was reduced by scavenging or inhibiting the expression of respiratory burst oxidase homologs (Rbohs) encoding ROS-producing plant NADPH oxidases, both of which catabolize SI. In addition, inhibition of pollen germination and growth on the stigma by increased ROS was similar to the expression of the self-incompatibility response. Interestingly, the process of pollination is similar to the defense mechanism induced by pathogenic bacteria invasion [35]. The cell wall will form defensive appositions called papillae when pathogens invade [65]. Additional involvement of ROS in papilla formation is that that they promote papilla sclerosis through cross-linking cell wall components that intoxicate pathogens and induce defense-related gene expression [66,67]. This suggests that the roles of proteins involved in these biological processes may be interrelated. The main biological function of the HMP family is to transport metal ions, but HMPs also act in flower development [3]. Based on RNA-seq data, we selected four *BrHMPs* as candidate genes that show elevated expression in self-incompatible pollination and decreased expression in compatible pollination. Variations in expression were further verified with qRT-PCR. *BrHMP36* expression started to increase significantly after 5 min of self-incompatible pollination and showed a decreasing trend at 5, 10, and 20 min of compatible pollination compared with that of the control (Figure 10). We predicted TL29 (AT4G09010) as a functional interaction partner of BrHMP36 by PPI on homologous gene AT2G36950 (Figure 11D). TL29 might participate in the defense mechanism of plants versus ROS and has therefore been renamed ascorbate peroxidase 4 (APX4) [53]. APX, like other ascorbate peroxidases, is a protein that protects plants from oxidative damage by transferring electrons from ascorbic acid and promoting the degradation of reactive oxygen species in chloroplasts [68]. This strengthens the likelihood that BrHMP36 may help regulate ROS produced by self- incompatibility.

In Brassica, self-incompatibility occurs by preventing pollen hydration, changes in metabolism, or by restricting the growth of pollen tubes into the stigma to reject self-incompatible pollen [69,70,71]. Excitingly, another prediction function partner of the homologous gene AT2G36950 of BrHMP36 is CHX18 (cation/H+ exchanger 18) (Figure 11D). The flowering plant genome encodes multiple cation/H+ exchangers (CHXs). The membrane transporters AtCHX17, AtCHX18, and AtCHX19 regulate K^+^ and pH homeostasis, and a triple mutant of these genes exhibits disordered reticulate morphology of pollen walls, indicating a defect in pollen wall formation during male gametophyte development [72] The walls of the pollen first touch the stigma of the pistil during pollination to determine if the pollen is adhering to the stigma [73]. The pollen wall is implicated in pollen adhesion, hydration, germination, pollen–stigma recognition and interactions [74], and the polar growth of the pollen tube [72] Therefore, we speculate that CHX18 may affect the hydration of pollen and the pollen–stigma interaction by altering pollen wall formation to affect self-incompatibility.

Protein phosphorylation is one of the main mechanisms regulating stress signal transduction in plants. In the results of the gene ontology analysis (Figure 5), the enriched molecular functions are mainly related to the binding and conjugation of metal ions, ATPase activity, and coupling across the membrane. In addition, several BrHMPs contain HAD domains that affect protein phosphorylation. The above results imply that phosphorylation modifications may occur in BrHMPs. We predicted that BrHMPs have a large number of phosphorylation sites (Figure 12). Related studies have demonstrated that in *Arabidopsis*, MYC2 is a master regulator of JA and its interaction with ABA, SA, and GAs and the growth hormone signaling pathway [75]. Importantly, MYC phosphorylation promotes the hydrolysis of coupled proteins and plays a key role in plant immunity [76]. We speculate that BrHMP may be involved in hormone signaling pathways such as JA to regulate the phosphorylation of coupled proteins in plants in response to abiotic stress.

## 5. Conclusions

Collectively, we identified 85 HMP genes in the *Brassica rapa* genome and performed comprehensive analysis of sequence features, *cis*-elements, and expression profiles of different tissues and plants under abiotic stress or after pollen–stigma interactions. We concluded that *BrHMP14, BrHMP16, BrHMP32, BrHMP41*, and *BrHMP42* likely regulate heavy metal stress tolerance; *BrHMP06* and *BrHMP78* are likely involved in abiotic stress tolerance; and *BrHMP36* functions in pollen–pistil interactions.

## Figures and Tables

**Figure 1 cells-12-01096-f001:**
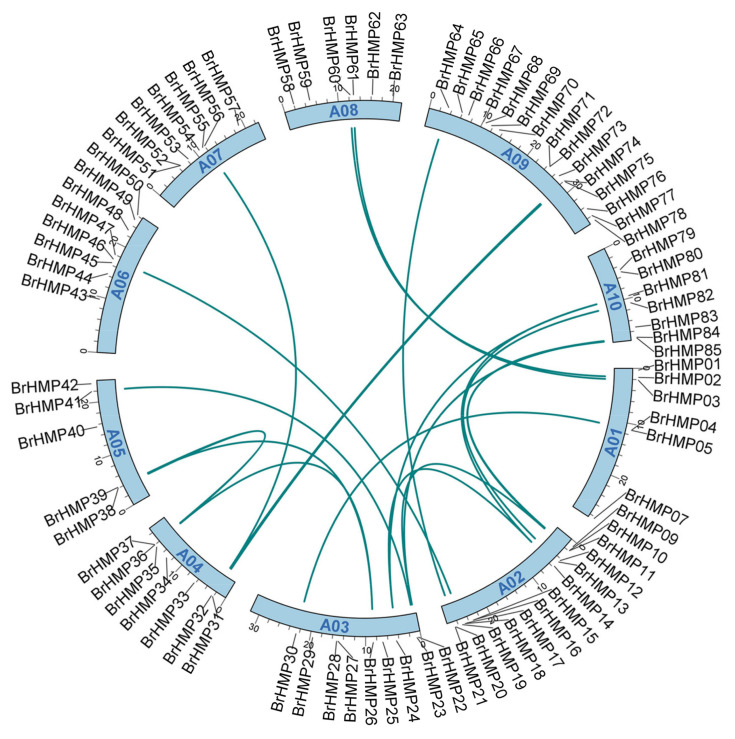
Chromosome localization and duplication events of *BrHMP* genes. The green lines indicate segmentally duplicated gene pairs.

**Figure 2 cells-12-01096-f002:**
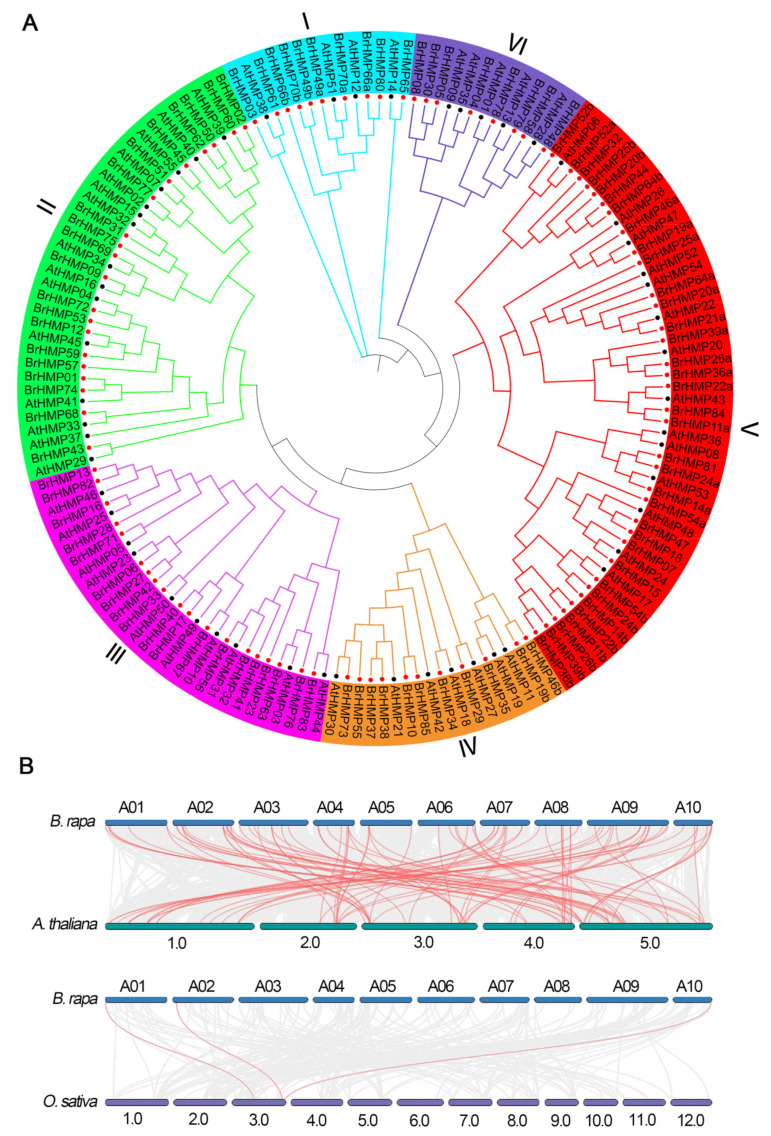
Phylogenetic tree and collinearity analysis of *BrHMPs*. (**A**) Phylogenetic tree of the *HMPs.* (**B**) Collinearity analysis of *HMP* genes in *B. rapa*, *A. thaliana*, and *O. sativa*. Homologous HMP gene pairs between species are linked by red lines.

**Figure 3 cells-12-01096-f003:**
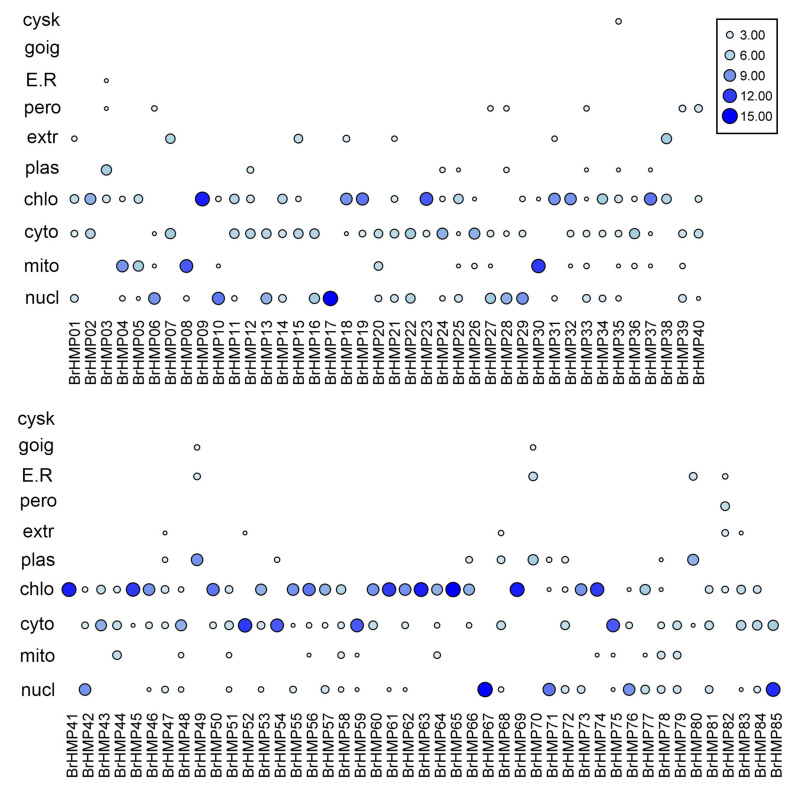
Subcellular localization analysis of BrHMPs. The color and the size of the circle indicate the reliability of the prediction. Nucl (nucleus), mito (mitochondria), cyto (cytoplasm), chlo (chloroplasts), plas (plasmids), extr (extracellular), pero (peroxisome), E.R (endoplasmic reticulum), goig (golgi apparatus), cysk (cytoskeleton).

**Figure 4 cells-12-01096-f004:**
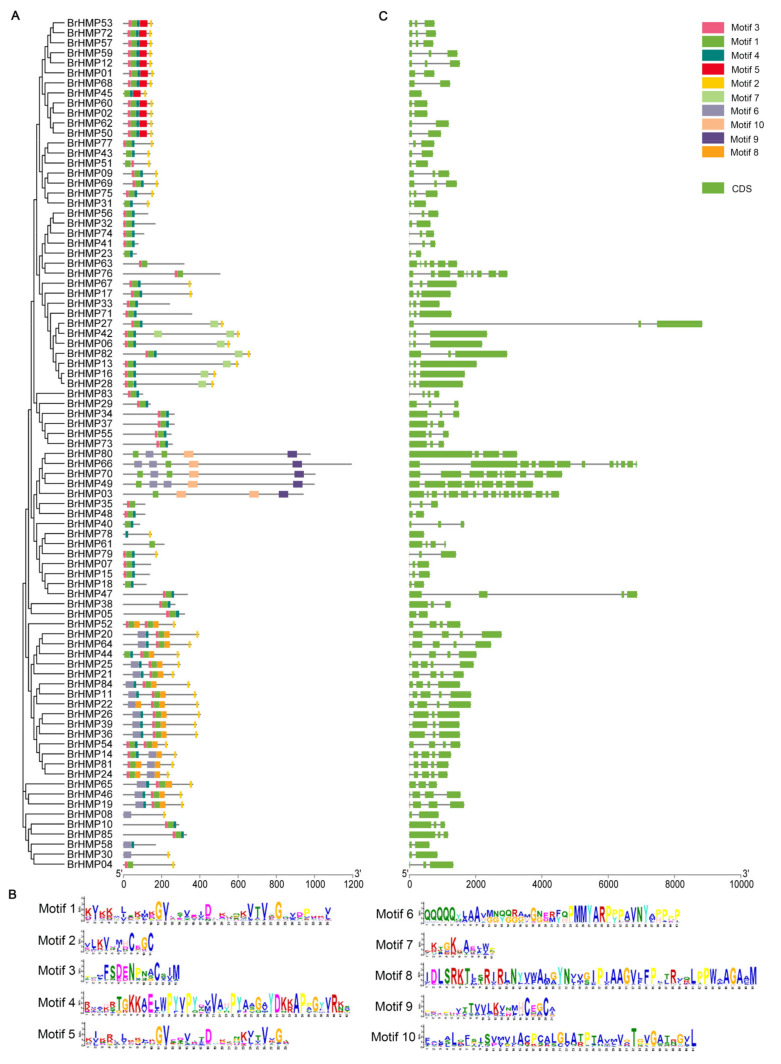
Gene structure and conserved motifs of BrHMPs. (**A**) Composition and distribution of conserved motifs. (**B**) Exon–intron structure of BrHMPs. Green boxes are exons, and lines connecting exons and introns are introns. (**C**) Ten conserved motif sequences of BrHMPs.

**Figure 5 cells-12-01096-f005:**
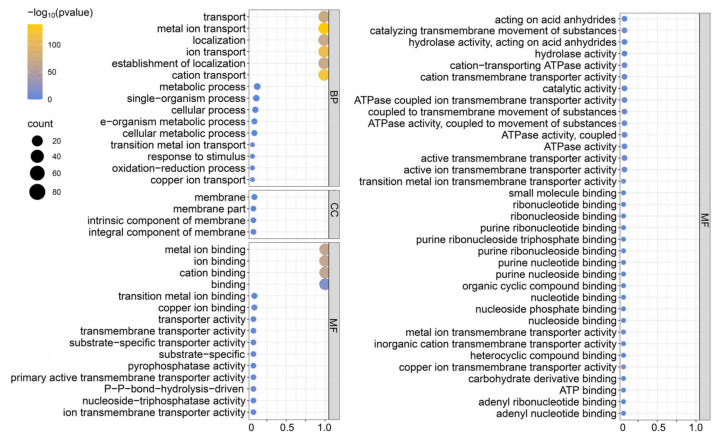
Gene Ontology analysis of *BrHMP* genes with enrichment in MF (molecular function), BP (biological process), and CC (cellular component) categories. The color and the size of the circles indicate the reliability of the prediction results.

**Figure 6 cells-12-01096-f006:**
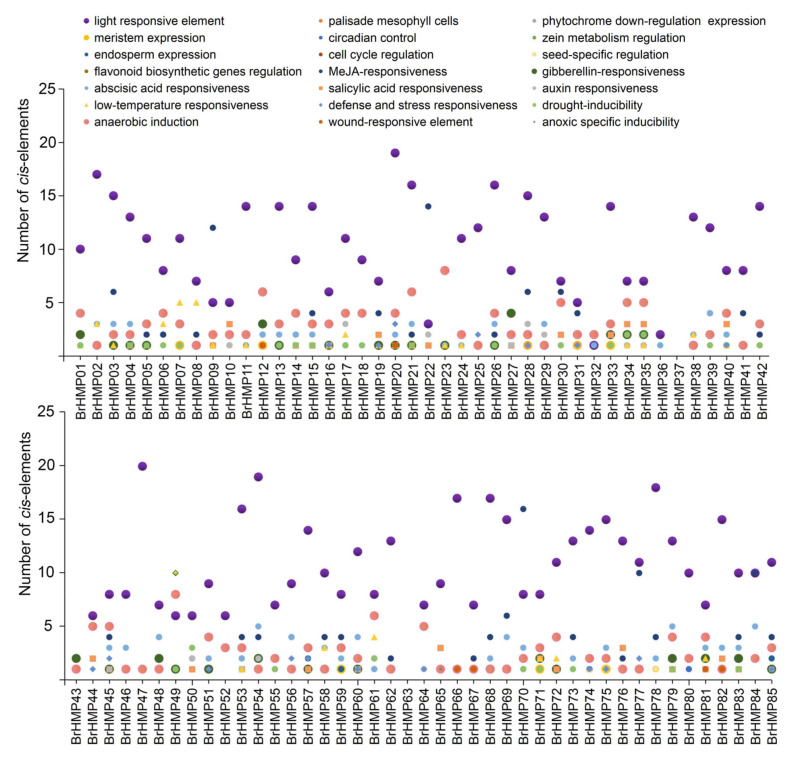
Analysis of *cis*-elements in promoters of *HMP* genes in *B. rapa*.

**Figure 7 cells-12-01096-f007:**
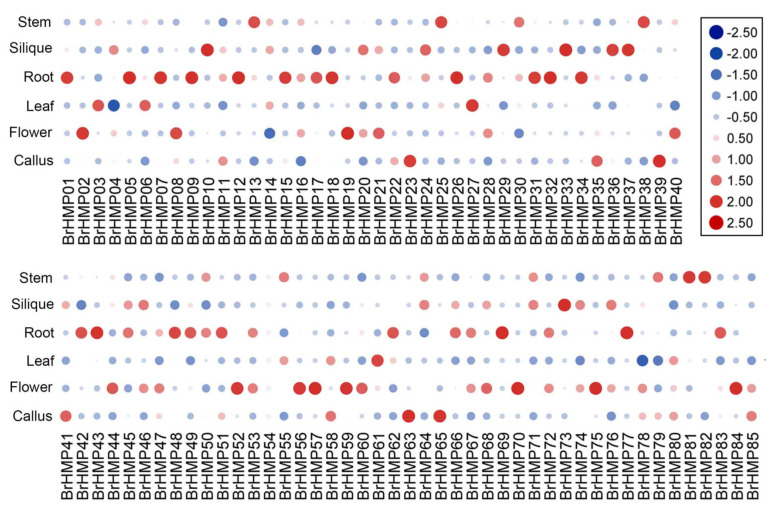
*BrHMPs* expression in different tissues of *B. rapa*. The abundance of each gene was measured by MPT.

**Figure 8 cells-12-01096-f008:**
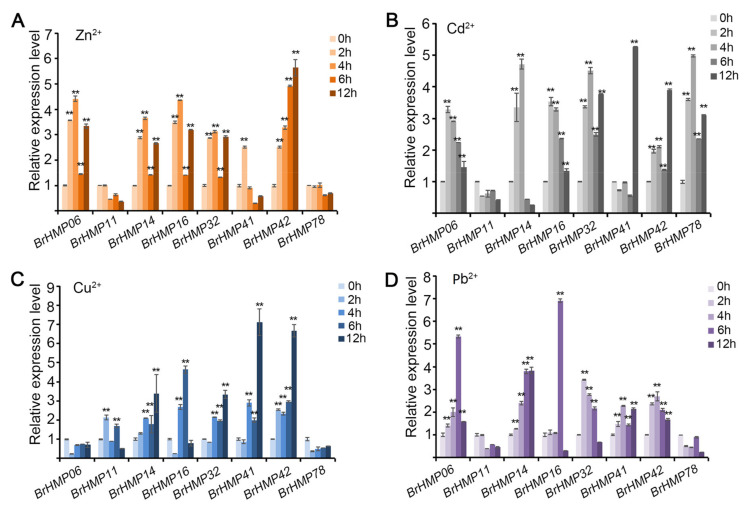
Expression profiles of *BrHMPs* in response to Zn^2+^ (**A**), Cd^2+^ (**B**), Cu^2+^ (**C**), and Pb^2+^ (**D**) stresses in *B. rapa* seedlings after treatment for 2 h, 4 h, 6 h, and 12 h, with 0 h as the control. Data are presented as means (±SD) of three biological replicates. ** *p* < 0.01, *t* test.

**Figure 9 cells-12-01096-f009:**
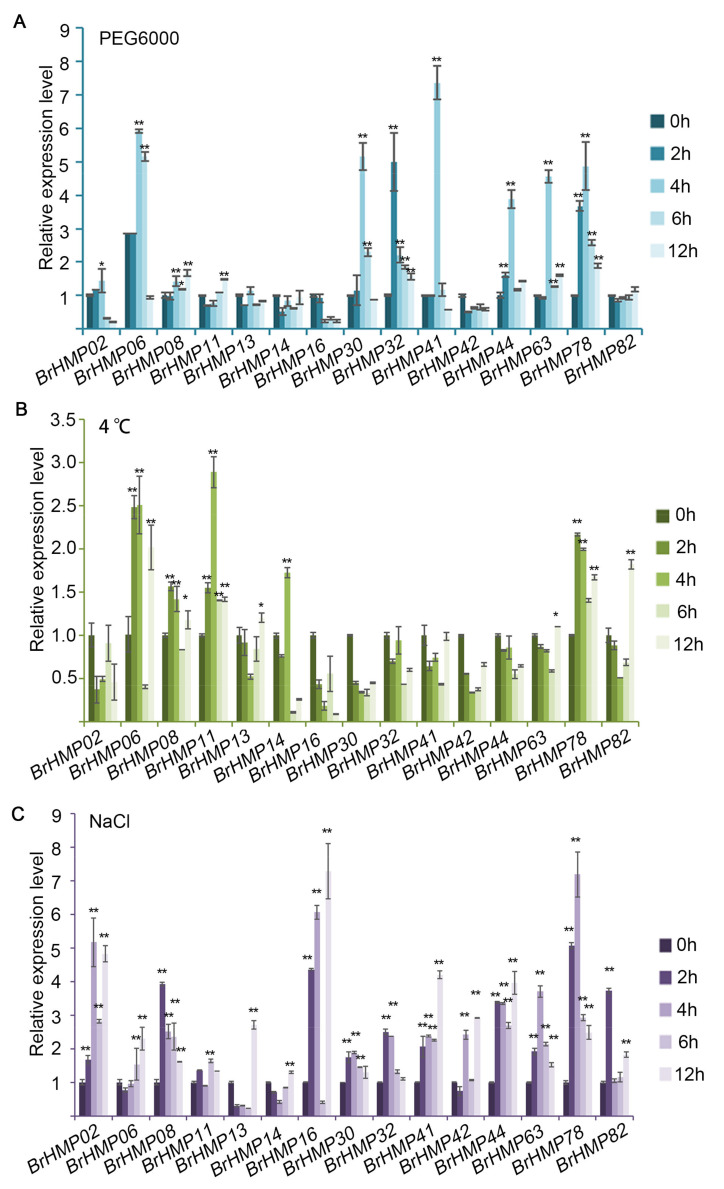
Analysis of the expression of 15 *BrHMP* genes in response to abiotic stress by qRT-PCR. (**A**) PEG6000 was used to simulate drought stress; (**B**) 4 °C was used to simulate cold stress; (**C**) NaCl was used to simulate salt stress. The experiment above was controlled with 0 h, and samples were tested after 2 h, 4 h, 6 h, and 12 h. The data are shown as mean (±SD), Student *t* test, * *p* < 0.05, ** *p* < 0.01.

**Figure 10 cells-12-01096-f010:**
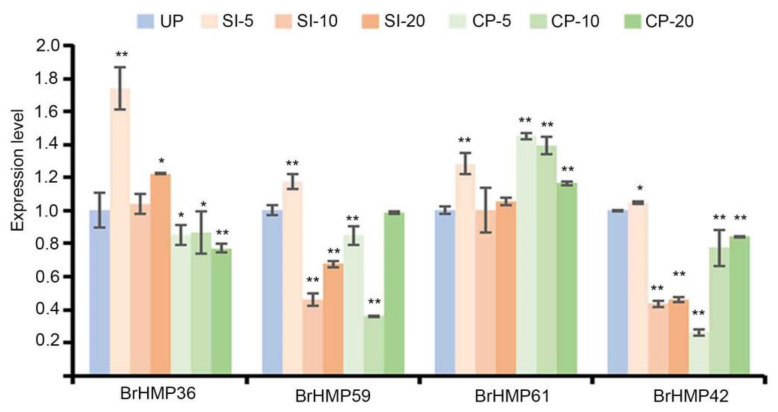
BrHMP gene expression profiles during SI and CP responses. Transcriptional level of four *BrHMP* genes at 5, 10, and 20 min after CP and SI pollination in the stigma. The experiments described above were carried out with UP as a control. UP (un-pollination), SI (self-incompatibility), CP (compatible pollination). The data are shown as mean (±SD), Student *t* test, * *p* < 0.05, ** *p* < 0.01.

**Figure 11 cells-12-01096-f011:**
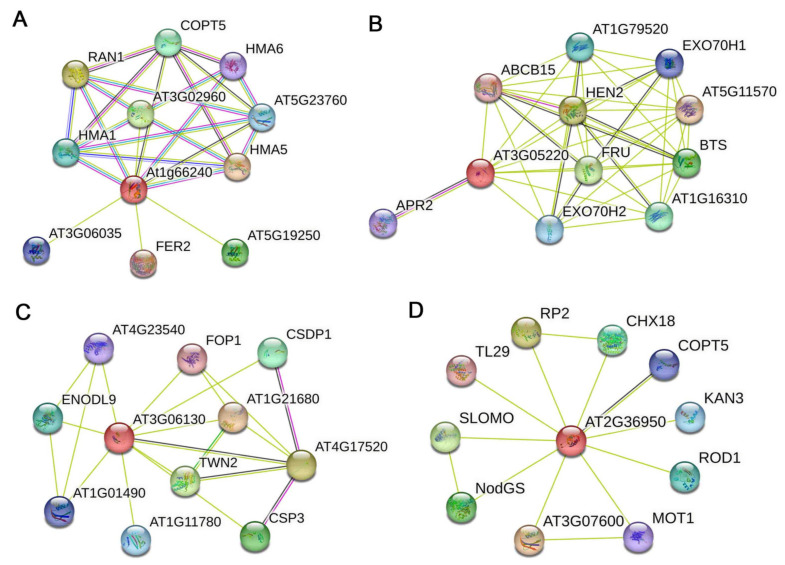
Network of protein–protein interactions of HMPs in *A. thaliana*. (**A**) AT1G66240 (*BrHMP41* homologous gene) PPIs. (**B**) AT3G05220 (*BrHMP06* and *BrHMP42* homologous gene) PPIs. (**C**) AT3G06130 (*BrHMP16* homologous gene) PPIs. (**D**) AT2G36950 (*BrHMP36* homologous gene) PPIs. Individual nodes indicate a protein, and interactions are linked by lines.

**Figure 12 cells-12-01096-f012:**
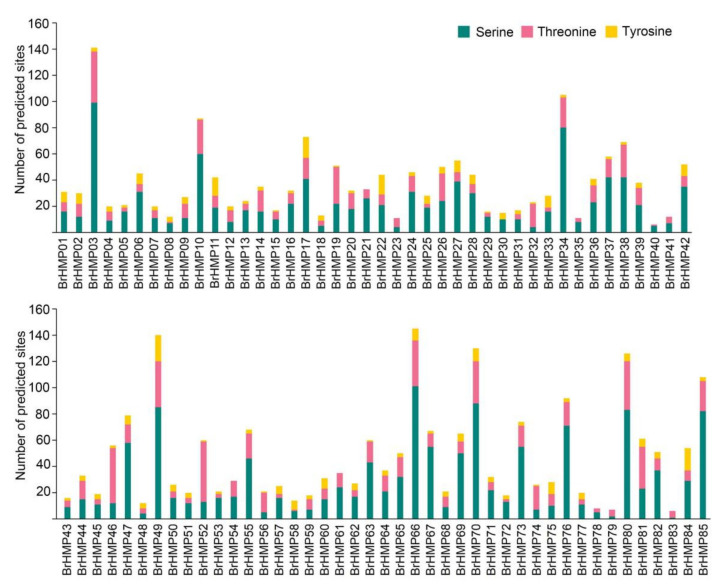
Distribution of predicted phosphorylation sites in BrHMP amino acid sequences.

## Data Availability

Data are contained within the article/Appendix A.

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
