# Peer review of "Comprehensive Analysis of BrHMPs Reveals Potential Roles in Abiotic Stress Tolerance and Pollen–Stigma Interaction in Brassica rapa"

_cells, 2023, doi:10.3390/cells12071096_

Round 1

Reviewer 1 Report

The manuscript entitled “Comprehensive Analysis of BrHMPs Reveal Potential Roles in Abiotic Stress Tolerance and Pollen-Stigma Interaction in Bras sica rapa” has identified the genes encoding heavy metal associated proteins (HMPs) in Brassica rapa genome. This study has also reported the structural properties of BrHMP genes as well as the expression analysis of some BrHMPs in response to various heavy metal and abiotic stress treatments. Finally, protein-protein interaction network analysis has been performed to more precisely reveal functions of BrHMPs in relation to pollen-stigma interaction and heavy metal response pathways. The manuscript is sound and publishable and represents some interesting results; however, I have some issues to arise:

-         - It is required to explain about the duplication events and paralogous copies of BrHMPs. This has been indicated in Figure 1, but there is no explanation in the text.

-          - It is required to report the method and results of phylogenetic analysis of BrHMPs.

-         - The reason and basis of selection of those eight BrHMP genes for expression analysis in response to heavy metal treatments is not clear. It has been mentioned “we selected eight BrHMPs with high expression in roots and leaves “. On the other hand, it has been reported that RNA extraction was performed from “samples of plants”. Therefore, I am not convinced about the basis of selection of the gens. Have the whole plants including roots been used for RNA extraction? Why have not roots and leaves been sampled separately? It is needed to be explained more clearley about this issue in the text.  

-          - Line 331: Self pollination (SI)? Self incompatability (SI) should be correct. Also, Cross pollination (CP) or Compatible pollination (CP) which has been mentioned in the abstract?

Author Response

Dear reviewer,

Best regards!

Reviewer 2 Report

In this work the authors identified different heavy metal associated proteins in Brassica rapa and gene expression studied under different abiotic conditions (osmotic stress and salinity) and heavy metals. Also, expression was determined during self-incompatibility and compatible pollination.

Using distinct bioinformatics methods and comparison with the homologues proteins in Arabidospis thaliana, protein-protein interaction and predicted phosphorylation were characterized.

The manuscript is well written and the used methodology is correct. Also, the interest of the manuscript is clear, since these proteins were not previously identified and characterized in a crop with agronomical value as B.rapa.

However it looks like a great bioinformatics analysis of genes was done but the functionality of the gene is clarify in the manuscript and there is a lack of discussion of many results.

There is a lack of correlation between the studied conditions for gene expression. The author should justify in the abstract and at the end of the introduction the multiple and variable studied conditions and the connection between them or any type of explanation about the selected conditions in order to characterize this family gene.

There is no discussion about the salinity and osmotic stress (PEG) results on gene expression

There is no discussion about the results obtained for different expression in distinct tissues, or subcellular location, cis elements or gene structure and protein phosphorylation predictions,

 weak conclusion as present form

Author Response

Dear reviewer,

Best regards!

Round 2

Reviewer 1 Report

The manuscript has been modified according to my comments and can be acceptable in the current form.